# RIFLE: Imputation and Robust Inference from Low Order Marginals

**Sina Baharlouei**                                                *baharlou@usc.edu*
*University of Southern California*

**Kelechi Ogudu**                                                  *kogudu@usc.edu*
*University of Southern California*

**Sze-chuan Suen**                                                 *ssuen@usc.edu*
*University of Southern California*

**Meisam Razaviyayn**                                              *razaviya@usc.edu*
*University of Southern California*

**Reviewed on OpenReview:** *https://openreview.net/forum?id=oud7NyOKQy*

## Abstract

The ubiquity of missing values in real-world datasets poses a challenge for statistical inference and can prevent similar datasets from being analyzed in the same study, precluding many existing datasets from being used for new analyses. While an extensive collection of packages and algorithms have been developed for data imputation, the overwhelming majority perform poorly if there are many missing values and low sample sizes, which are unfortunately common characteristics in empirical data. Such low-accuracy estimations adversely affect the performance of downstream statistical models. We develop a statistical inference framework for *regression and classification in the presence of missing data without imputation*. Our framework, RIFLE (Robust InFerence via Low-order moment Estimations), estimates low-order moments of the underlying data distribution with corresponding confidence intervals to learn a distributionally robust model. We specialize our framework to linear regression and normal discriminant analysis, and we provide convergence and performance guarantees. This framework can also be adapted to impute missing data. In numerical experiments, we compare RIFLE to several state-of-the-art approaches (including MICE, Amelia, MissForest, KNN-imputer, MIDA, and Mean Imputer) for imputation and inference in the presence of missing values. Our experiments demonstrate that RIFLE outperforms other benchmark algorithms when the percentage of missing values is high and/or when the number of data points is relatively small. RIFLE is publicly available at `https://github.com/optimization-for-data-driven-science/RIFLE`.

## 1 Introduction

Machine learning algorithms have shown promise when applied to various problems, including healthcare, finance, social data analysis, image processing, and speech recognition. However, this success mainly relied on the availability of large-scale, high-quality datasets, which may be scarce in many practical problems, especially in medical and health applications (Pedersen et al., 2017; Sterne et al., 2009; Beaulieu-Jones et al., 2018). Moreover, many experiments and datasets suffer from the small sample size in such applications. Despite the availability of a small number of data points in each study, an increasingly large number of datasets are publicly available. To fully and effectively utilize information on related research questions from diverse datasets, information across various datasets (e.g., different questionnaires from multiple hospitals with overlapping questions) must be combined in a reliable fashion.

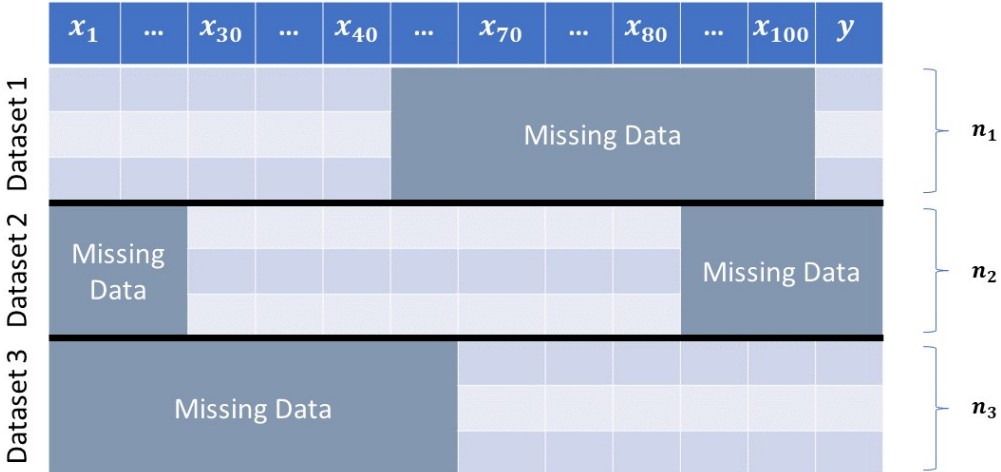

Figure 1: Consider the problem of predicting the trait $y$ from feature vector $(\mathbf{x}_1, \ldots, \mathbf{x}_{100})$. Suppose that we have access to three data sets: The first dataset includes the measurements of $(\mathbf{x}_1, \mathbf{x}_2, \ldots, \mathbf{x}_{40}, y)$ for $n_1$ individuals. The second dataset collects data from another $n_2$ individuals by measuring $(\mathbf{x}_{30}, \ldots, \mathbf{x}_{80})$ with no measurements of the target variable $y$ in it; and the third dataset contains the measurements from the variables $(\mathbf{x}_{70}, \ldots, \mathbf{x}_{100}, y)$ for $n_3$ number of individuals. How one should learn the predictor $\hat{y} = h(\mathbf{x}_1, \ldots, \mathbf{x}_{100})$ from these three datasets?

After integrating data from different studies, the obtained dataset can contain large blocks of missing values, as they may not share the same features (Figure 1).

There are three general approaches for handling missing values in statistical inference (classification and regression) tasks. A Naïve method is to remove the rows containing missing entries. However, such an approach is not an option when the percentage of missingness in a dataset is high. For instance, as demonstrated in Figure 1, the entire dataset will be discarded if we eliminate the rows with at least one missing entry.

The most common methodology for handling missing values in a learning task is to impute them in a preprocessing stage. The general idea behind data imputation is that the missing values can be predicted using the available data entries and correlated features. Imputation algorithms cover a wide range of methods, including imputing missing entries with the columns means Little & Rubin (2019, Chapter 3) (or median), least-square and linear regression-based methods (Raghunathan et al., 2001; Kim et al., 2005; Zhang et al., 2008; Cai et al., 2006; Buuren & Groothuis-Oudshoorn, 2010), matrix completion and expectation maximization approaches Dempster et al. (1977); Ghahramani & Jordan (1994); Honaker et al. (2011), KNN based (Troyanskaya et al., 2001), Tree based methods (Stekhoven & Bühlmann, 2012; Xia et al., 2017), and methods using different neural network structures. Appendix A presents a comprehensive review of these methods.

The imputation of data allows practitioners to run standard statistical algorithms requiring complete data. However, the prediction model's performance can be highly reliant on the accuracy of the imputer. High error rates in the prediction of missing values by the imputer can lead to the catastrophic performance of the downstream statistical methods executed on the imputed data.

Another class of methods for inference in the presence of missing values relies on robust optimization over the uncertainty sets on missing entries. Shivaswamy et al. (2006) and Xu et al. (2009) adopt robust optimization to learn the parameters of a support vector machine model. They consider uncertainty sets for the missing entries in the dataset and solve a min-max problem over those sets. The obtained classifiers are robust to the uncertainty of missing entries within the uncertainty regions. In contrast to the imputation-based approaches, the robust classification formulation does not carry the imputation error to the classification phase. However, finding appropriate intervals for each missing entry is challenging, and it is unclear how to determine the uncertainty range in many real datasets. Moreover, their proposed algorithms are limited to the SVM classifier.

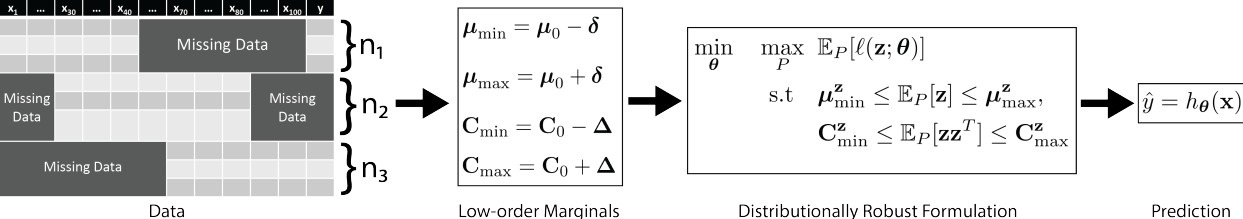

Figure 2: Prediction of the target variable without imputation. RIFLE estimates confidence intervals for low-order (first and second-order) marginals from the input data containing missing values. Then, it solves a distributionally robust problem over the set of all distributions whose low-order marginals are within the estimated confidence intervals.

In this paper, we propose RIFLE (Robust InFerence via Low-order moment Estimations) for the direct inference of a target variable based on a set of features containing missing values. The proposed framework does not require the data to be imputed in a pre-processing stage. However, it can also be used as a pre-processing tool for imputing data. The main idea of the proposed framework is to estimate the first and second-order moments of the data and their confidence intervals by bootstrapping on the available data matrix entries. Then, RIFLE finds the optimal parameters of the statistical model for the worst-case distribution with the low-order moments (mean and variance) within the estimated confidence intervals (See Figure 2). Compared to Shivaswamy et al. (2006); Xu et al. (2009), we estimate uncertainty regions for the low-order marginals using the Bootstrap technique. Furthermore, our framework is not restricted to any particular machine learning model, such as support vector machines (Xu et al., 2009).

**Contributions:** Our main contributions are as follows:

1. We present a distributionally robust optimization framework over the low-order marginals of the training data distribution for inference in the presence of missing values. The proposed framework does not require data imputation as a pre-processing stage. In Section 3 and Section 4, we specialize the framework to ridge regression and classification models as two case studies respectively. The proposed framework provides a novel strategy for inference in the presence of missing data, especially for datasets with large proportions of missing values.

2. We provide theoretical convergence guarantees and the iteration complexity analysis of the presented algorithms for robust formulations of ridge linear regression and normal discriminant analysis. Moreover, we show the consistency of the prediction under mild assumptions and analyze the asymptotic statistical properties of the solutions found by the algorithms.

3. While the robust inference framework is primarily designed for direct statistical inference in the presence of missing values without performing data imputation, it can also be adopted as an imputation tool. To demonstrate the quality of the proposed imputer, we compare its performance with several widely-used imputation packages such as MICE (Buuren & Groothuis-Oudshoorn, 2010), Amelia (Honaker et al., 2011), MissForest (Stekhoven & Bühlmann, 2012), KNN-Imputer (Troyanskaya et al., 2001), MIDA (Gondara & Wang, 2018), GAIN (Yoon et al., 2018) on real and synthetic datasets. Generally speaking, our method outperforms all of the mentioned packages when the number of missing entries is large.

## 2 Robust Inference via Estimating Low-order Moments

RIFLE is based on a distributionally robust optimization (DRO) framework over low-order marginals. Assume that $(\mathbf{x}, y) \in \mathbb{R}^d \times \mathbb{R}$ follows a joint probability distribution $P^*$. A standard approach for predicting the target variable $y$ given the input vector $\mathbf{x}$ is to find the parameter $\boldsymbol{\theta}$ that minimizes the population risk with respect to a given loss function $\ell$:

$$\min_{\boldsymbol{\theta}} \quad \mathbb{E}_{(\mathbf{x}, y) \sim P^*}\Big[\ell\Big(\mathbf{x}, y; \boldsymbol{\theta}\Big)\Big]. \tag{1}$$

Since the underlying distribution of data is rarely available in practice, the above problem cannot be directly solved. The most common approach for approximating (1) is to minimize the empirical risk with respect to

$n$ given i.i.d samples $(\mathbf{x}_1, y_1), \ldots, (\mathbf{x}_n, y_n)$ drawn from the joint distribution $P^*$:

$$\min_{\boldsymbol{\theta}} \quad \frac{1}{n} \sum_{i=1}^n \ell(\mathbf{x}_i, y_i; \boldsymbol{\theta}).$$

The above empirical risk formulation assumes that all entries of $\mathbf{x}_i$ and $y_i$ are available. Thus, to utilize the empirical risk minimization (ERM) framework in the presence of missing values, one can either remove or impute the missing data points in a pre-processing stage. Training via robust optimization is a natural alternative in the presence of missing data. Shivaswamy et al. (2006); Xu et al. (2009) suggest the following optimization problem that minimizes the loss function for the worst-case scenario over the defined uncertainty sets per data points:

$$\min_{\boldsymbol{\theta}} \quad \max_{\{\boldsymbol{\delta}_i \in \mathcal{N}_i\}_{i=1}^n} \quad \frac{1}{n} \sum_{i=1}^n \ell(\mathbf{x}_i - \boldsymbol{\delta}_i, y_i; \boldsymbol{\theta}), \tag{2}$$

where $\mathcal{N}_i$ represents the uncertainty region of data point $i$. Shivaswamy et al. (2006) obtains the uncertainty sets by assuming a known distribution on the missing entries of datasets. The main issue in their approach is that the constraints defined on data points are totally uncorrelated. Xu et al. (2009) on the other hand defines $\mathcal{N}_i$ as a "box" constraint around the data point $i$ such that they can be linearly correlated. For this specific case, they show that solving the corresponding robust optimization problem is equivalent to minimizing a regularized reformulation of the original loss function. Such an approach has several limitations: First, it can only handle a few special cases (SVM loss with linearly correlated perturbations on data points). Furthermore, Xu et al. (2009) is primarily designed for handling outliers and contaminated data. Thus, they do not offer any mechanism for the initial estimation of $\mathbf{x}_i$ when several vector entries are missing. In this work, we instead take a *distributionally robust* approach by considering uncertainty on the data distribution instead of defining an uncertainty set for each data point. In particular, we aim to fit the best parameters of a statistical learning model for the worst distribution in a given uncertainty set by solving the following:

$$\min_{\boldsymbol{\theta}} \quad \max_{P \in \mathcal{P}} \quad \mathbb{E}_{(\mathbf{x}, y) \sim P}[\ell(\mathbf{x}, y; \boldsymbol{\theta})], \tag{3}$$

where $\mathcal{P}$ is an uncertainty set over the underlying distribution of data. A key observation is that defining the uncertainty set $\mathcal{P}$ in (3) is easier and computationally more efficient than defining the uncertainty sets $\{\mathcal{N}_i\}_{i=1}^n$ in (2). In particular, the uncertainty set $\mathcal{P}$ can be obtained naturally by estimating low-order moments of data distribution using only available entries. To explain this idea and to simplify the notations, let $\mathbf{z} = (\mathbf{x}, y)$, $\bar{\boldsymbol{\mu}}^{\mathbf{z}} \triangleq \mathbb{E}[\mathbf{z}]$, and $\bar{\mathbf{C}}^{\mathbf{z}} \triangleq \mathbb{E}[\mathbf{z}\mathbf{z}^T]$. While $\bar{\boldsymbol{\mu}}^{\mathbf{z}}$ and $\bar{\mathbf{C}}^{\mathbf{z}}$ are typically not known exactly, one can estimate them (within certain confidence intervals) from the available data by simply ignoring missing entries (assuming the missing value pattern is completely at random, e.g., MCAR). Moreover, we can estimate the confidence intervals via bootstrapping. Particularly, we can estimate $\boldsymbol{\mu}_{\min}^{\mathbf{z}}, \boldsymbol{\mu}_{\max}^{\mathbf{z}}, \mathbf{C}_{\min}^{\mathbf{z}}$, and $\mathbf{C}_{\max}^{\mathbf{z}}$ from data such that $\boldsymbol{\mu}_{\min}^{\mathbf{z}} \leq \bar{\boldsymbol{\mu}}^{\mathbf{z}} \leq \boldsymbol{\mu}_{\max}^{\mathbf{z}}$ and $\mathbf{C}_{\min}^{\mathbf{z}} \leq \bar{\mathbf{C}}^{\mathbf{z}} \leq \mathbf{C}_{\max}^{\mathbf{z}}$ with high probability (where the inequalities for matrices and vectors denote component-wise relations). In Appendix B, we show how a bootstrapping strategy can be used to obtain the confidence intervals described above. Given these estimated confidence intervals from data, (3) can be reformulated as

$$\begin{aligned} \min_{\boldsymbol{\theta}} \quad & \max_{P} \quad \mathbb{E}_P[\ell(\mathbf{z}; \boldsymbol{\theta})] \\ & \text{s.t.} \quad \boldsymbol{\mu}_{\min}^{\mathbf{z}} \leq \mathbb{E}_P[\mathbf{z}] \leq \boldsymbol{\mu}_{\max}^{\mathbf{z}}, \\ & \quad \mathbf{C}_{\min}^{\mathbf{z}} \leq \mathbb{E}_P[\mathbf{z}\mathbf{z}^T] \leq \mathbf{C}_{\max}^{\mathbf{z}}. \end{aligned} \tag{4}$$

Gao & Kleywegt (2017) utilize the distributionally robust optimization as (3) over the set of positive semi-definite (PSD) cones for robust inference under uncertainty. While their formulation considers $\ell_2$ balls for the constraints on low order moments of the data, we use $\ell_\infty$ constraints that are computationally more natural in the presence of missing entries when combined with bootstrapping. Furthermore, while it can be applied to general convex losses, their method relies on the ellipsoid and the existence of oracles for performing the steps of the ellipsoid method, which is not applicable in modern high-dimensional problems. Moreover, they assume concavity in data (the existence of some oracle to return the worst-case data points) that is

practically unavailable even in convex loss functions (including linear regression and normal discriminant analysis studied in our work).

In Section 3, we study the proposed distributionally robust framework described in (4) for the ridge linear regression. We design efficient first-order convergent algorithms to solve the problem and show how we can use the algorithms for both inference and imputation in the presence of missing values. Further, in Appendix F, we study the proposed distributionally robust framework for the classification problems under the normality assumption of features. In particular, we show how Framework (4) can be specialized to the robust normal discriminant analysis in the presence of missing values.

## 3   Robust Linear Regression in the Presence of Missing Values

Let us specialize our framework to the ridge linear regression model. In the absence of missing data, ridge regression finds optimal regressor parameter $\boldsymbol{\theta}$ by solving

$$\min_{\boldsymbol{\theta}} \quad \|\mathbf{X}\boldsymbol{\theta} - \mathbf{y}\|_2^2 + \lambda\|\boldsymbol{\theta}\|_2^2,$$

or equivalently by solving:

$$\min_{\boldsymbol{\theta}} \quad \boldsymbol{\theta}^T\mathbf{X}^T\mathbf{X}\boldsymbol{\theta} - 2\boldsymbol{\theta}^T\mathbf{X}^T\mathbf{y} + \lambda\|\boldsymbol{\theta}\|_2^2. \tag{5}$$

Thus, having the second-order moments of the data $\mathbf{C} = \mathbf{X}^T\mathbf{X}$ and $\mathbf{b} = \mathbf{X}^T\mathbf{y}$ is sufficient for finding the optimal solution. In other words, it suffices to compute the inner product of any two column vectors $\mathbf{a}_i$, $\mathbf{a}_j$ of $\mathbf{X}$, and the inner product of any column $\mathbf{a}_i$ of $\mathbf{X}$ with vector $\mathbf{y}$. Since the matrix $\mathbf{X}$ and vector $\mathbf{y}$ are not fully observed due to the existence of missing values, one can use the available data (see (24) for details) to compute the point estimators $\mathbf{C}_0$ and $\mathbf{b}_0$. These point estimators can be highly inaccurate, especially when the number of non-missing rows for two given columns is small. In addition, if the pattern of missing entries does not follow the MCAR assumption, the point estimators are not unbiased estimators of $\mathbf{C}$ and $\mathbf{b}$.

### 3.1   A Distributionally Robust Formulation of Linear Regression

As we mentioned above, to solve the linear regression problem, we only need to estimate the second-order moments of the data ($\mathbf{X}^T\mathbf{X}$ and $\mathbf{X}^T\mathbf{y}$). Thus, the distributionally robust formulation described in (4) is equivalent to the following optimization problem for the linear regression model:

$$
\begin{aligned}
\min_{\boldsymbol{\theta}} \quad \max_{\mathbf{C},\mathbf{b}} \quad & \boldsymbol{\theta}^T\mathbf{C}\boldsymbol{\theta} - 2\mathbf{b}^T\boldsymbol{\theta} + \lambda\|\boldsymbol{\theta}\|_2^2 \\
\text{s.t.} \quad & \mathbf{C}_0 - c\boldsymbol{\Delta} \le \mathbf{C} \le \mathbf{C}_0 + c\boldsymbol{\Delta}, \\
& \mathbf{b}_0 - c\boldsymbol{\delta} \le \mathbf{b} \le \mathbf{b}_0 + c\boldsymbol{\delta}, \\
& \mathbf{C} \succeq 0,
\end{aligned}
\tag{6}
$$

where the last constraint guarantees that the covariance matrix is positive and semi-definite. We dicuss the procedure of estimating the confidence intervals ($\mathbf{b}_0, \mathbf{C}_0, \boldsymbol{\delta}$, and $\boldsymbol{\Delta}$) in Appendix B.

### 3.2   RIFLE for Ridge Linear Regression

Since the objective function in (6) is convex in $\boldsymbol{\theta}$ (ridge regression) and concave in $\mathbf{b}$ and $\mathbf{C}$ (linear), the minimization and maximization sub-problems are interchangeable (Sion et al., 1958). Thus, we can equivalently rewrite Problem (6) as:

$$
\begin{aligned}
\max_{\mathbf{C},\mathbf{b}} \quad & g(\mathbf{C},\mathbf{b}) \\
\text{s.t.} \quad & \mathbf{C}_0 - c\boldsymbol{\Delta} \le \mathbf{C} \le \mathbf{C}_0 + c\boldsymbol{\Delta}, \\
& \mathbf{b}_0 - c\boldsymbol{\delta} \le \mathbf{b} \le \mathbf{b}_0 + c\boldsymbol{\delta}, \\
& \mathbf{C} \succeq 0,
\end{aligned}
\tag{7}
$$

where $g(\mathbf{b}, \mathbf{C}) = \min_{\boldsymbol{\theta}} \boldsymbol{\theta}^T \mathbf{C} \boldsymbol{\theta} - 2\mathbf{b}^T \boldsymbol{\theta} + \lambda \|\boldsymbol{\theta}\|^2$. Function $g$ can be computed in closed-form given any pair of $(\mathbf{C}, \mathbf{b})$ by setting $\boldsymbol{\theta} = (\mathbf{C} + \lambda \mathbf{I})^{-1} \mathbf{b}$. Thus, using Danskin's Theorem (Danskin, 2012), we can apply projected gradient ascent to function $g$ to find an optimal solution of (7) as described in Algorithm 1. At each iteration of the algorithm, we first perform one step of projected gradient ascent on matrix $\mathbf{C}$ and vector $\mathbf{b}$; then we update $\boldsymbol{\theta}$ in closed-form for the obtained $\mathbf{C}$ and $\mathbf{b}$. We initialize $\mathbf{C}$ and $\mathbf{b}$ using entriwise point estimation on the available rows (see Equation (24) in Appendix B). The projection of $\mathbf{b}$ to the box

---

**Algorithm 1** RIFLE for Ridge Linear Regression in the Presence of Missing Values

1: **Input:** $\mathbf{C}_0, \mathbf{b}_0, \boldsymbol{\Delta}, \boldsymbol{\delta}, T$
2: **Initialize:** $\mathbf{C} = \mathbf{C}_0, \mathbf{b} = \mathbf{b}_0$.
3: **for** $i = 1, \ldots, T$ **do**
4:     Update $\mathbf{C} = \Pi_{\boldsymbol{\Delta}+} \left[ \mathbf{C} + \alpha \boldsymbol{\theta} \boldsymbol{\theta}^T \right]$
5:     Update $\mathbf{b} = \Pi_{\boldsymbol{\delta}} (\mathbf{b} - 2\alpha \boldsymbol{\theta})$
6:     Set $\boldsymbol{\theta} = (\mathbf{C} + \lambda \mathbf{I})^{-1} \mathbf{b}$

---

constraint $\mathbf{b}_0 - c\boldsymbol{\delta} \leq \mathbf{b} \leq \mathbf{b}_0 + c\boldsymbol{\delta}$ can be done entriwise and has the following closed-form

$$\Pi_{\delta}(\mathbf{b}_i) = \begin{cases} \mathbf{b}_i & \text{if} \quad \mathbf{b}_{0i} - c\boldsymbol{\delta}_i \leq \mathbf{b}_i \leq \mathbf{b}_{0i} + c\boldsymbol{\delta}_i, \\ \mathbf{b}_{0i} - c\boldsymbol{\delta}_i & \text{if} \quad \mathbf{b}_i < \mathbf{b}_{0i} - c\boldsymbol{\delta}_i, \\ \mathbf{b}_{0i} + c\boldsymbol{\delta}_i & \text{if} \quad \mathbf{b}_{0i} + c\boldsymbol{\delta}_i < \mathbf{b}_i. \end{cases}$$

**Theorem 1.** *Let $(\tilde{\boldsymbol{\theta}}, \tilde{\mathbf{C}}, \tilde{\mathbf{b}})$ be the optimal solution of (6), $\boldsymbol{\theta}^*(\mathbf{b}, \mathbf{C}) = \arg\min_{\boldsymbol{\theta}} \boldsymbol{\theta}^T \mathbf{C} \boldsymbol{\theta} - 2\mathbf{b}^T \boldsymbol{\theta} + \lambda \|\boldsymbol{\theta}\|^2$, and $D = \|\mathbf{C}_0 - \tilde{\mathbf{C}}\|_F^2 + \|\mathbf{b}_0 - \tilde{\mathbf{b}}\|_2^2$. Assume that for any given $\mathbf{b}$ and $\mathbf{C}$, within the uncertainty (constraint) sets described in (6), $\|\boldsymbol{\theta}^*(\mathbf{b}, \mathbf{C})\| \leq \tau$. Then Algorithm 1 computes an $\epsilon$-optimal solution of the objective function in (7) in $\mathcal{O}\left(\frac{D(\tau+1)^2}{\lambda \epsilon}\right)$ iterations.*

*Proof.* The proof is relegated to Appendix H. $\qquad\qquad\qquad\qquad\qquad\qquad\qquad\qquad\qquad\qquad\qquad\qquad\qquad$ $\square$

In Appendix C, we show how using the acceleration method of Nesterov can improve the convergence rate of Algorithm 1 to $\mathcal{O}\left(\sqrt{\frac{D(\tau+1)^2}{\epsilon \lambda}}\right)$. A technical issue of Algorithm 1 and its accelerated version presented in Appendix C is that projection of $\mathbf{C}$ to the intersection of box constraints and the set of positive semidefinite matrices ($\Pi_{\boldsymbol{\Delta}+} [\mathbf{C}]$) is challenging and cannot be done in closed-form. In the implementation of Algorithm 1, we relax the problem by removing the PSD constraint on $\mathbf{C}$ to avoid this complexity and time-consuming singular value decomposition at each iteration. This relaxation does not drastically change the algorithm's performance, as our experiments show in Section 5. A more systematic approach is to write the dual problem of the maximization problem and handle the resulting constrained minimization problem with the Alternating Direction Method of Multipliers (ADMM). The detailed procedure of such an approach can be found in Appendix D. All these algorithms are provably convergent to the optimal points of Problem (6). In addition to theoretical convergence, we have numerically evaluated the convergence of resulting algorithms in Appendix K. Further, the proposed algorithms are **consistent**, as discussed in Appendix J.

### 3.3 Performance Guarantees for RIFLE

Thus far, we have discussed how to efficiently solve the robust linear regression problem in the presence of missing values. A natural question in this context is the statistical performance of the obtained optimal solution in the previous section on the unseen test data points. Theorem 2 answers this question from two perspectives: Assuming that the missing values are distributed completely at random, our estimators are consistent. Moreover, for the finite case, Theorem 2 part (b) states that with the proper choice of confidence intervals, with high probability, the test loss of the obtained solution is bounded by the training loss of the estimator. Note that the results regarding the performance of the robust estimator generally hold for MCAR missing pattern. However, we perform several experiments on datasets with MNAR patterns to show how RIFLE works in practice on such datasets in Section 5.

**Theorem 2.** *Assume the data domain is bounded and that the missing pattern of the data follows MCAR. Let $\mathbf{X}^{n \times d}$, $\mathbf{y}$ be the training data drawn i.i.d. from the ground-truth distribution $P^*$ with low-order moments $\mathbf{C}^*$ and $\mathbf{b}^*$. Further, assume that each entry of $\mathbf{X}$ and $\mathbf{y}$ is missing with probability $p < 1$. Let $(\tilde{\boldsymbol{\theta}}_n, \tilde{\mathbf{C}}_n, \tilde{\mathbf{b}}_n)$ be the solution of Problem* (6).

*(a) Consistency of the Covariance Estimator: As the number of data points goes to infinity, the estimated low-order marginals converge to the ground-truth values, almost surely. More precisely,*

$$\lim_{n \to \infty} \tilde{\mathbf{C}}_n = \mathbb{E}_{P^*}[\mathbf{x}\mathbf{x}^T], \quad a.s., \tag{8}$$

$$\lim_{n \to \infty} \tilde{\mathbf{b}}_n = \mathbb{E}_{P^*}[\mathbf{x}y], \quad a.s. \tag{9}$$

*(b) Defining*

$$L_{\text{train}}(\tilde{\boldsymbol{\theta}}_n) = \tilde{\boldsymbol{\theta}}_n^T \tilde{\mathbf{C}}_n \tilde{\boldsymbol{\theta}}_n - 2\tilde{\mathbf{b}}_n \tilde{\boldsymbol{\theta}}_n + \lambda \|\tilde{\boldsymbol{\theta}}_n\|_2^2$$

$$L_{\text{test}}(\tilde{\boldsymbol{\theta}}_n) = \tilde{\boldsymbol{\theta}}_n^T \mathbf{C}^* \tilde{\boldsymbol{\theta}}_n - 2\mathbf{b}^{*T} \tilde{\boldsymbol{\theta}}_n + \lambda \|\tilde{\boldsymbol{\theta}}_n\|_2^2,$$

*where $\mathbf{C}^* = \mathbb{E}_{(\mathbf{x},y) \sim P^*}[\mathbf{x}\mathbf{x}^T]$ and $\mathbf{b}^* = \mathbb{E}_{(\mathbf{x},y) \sim P^*}[\mathbf{x}y]$ are the ground-truth second-order moments. Given $V = \max_{i,j} \text{Var}(X_i X_j)$ (maximum variance of pairwise feature products), with the probability of at least $1 - \frac{d^2 V}{2c^2 \boldsymbol{\Delta}^2 n(1-p)}$, we have:*

$$L_{\text{test}}(\tilde{\boldsymbol{\theta}}) \leq L_{\text{train}}(\tilde{\boldsymbol{\theta}}), \tag{10}$$

*where $\Delta = \min\{\Delta_{ij}\}$ and $c$ is the hyper-parameter for controlling the size of the confidence intervals as presented in* (6)

*Proof.* The proof is relegated to Appendix H. □

## 3.4 Imputation of Missing Values and Going Beyond Linear Regression

RIFLE can be used for imputing missing data. To this end, we impute different features of a given dataset independently. More precisely, to impute each feature containing missing values, we consider it as a target variable $\mathbf{y}$ and the rest of the features as the input $\mathbf{X}$ in our methodology. Then, we train a model to predict the feature $\mathbf{y}$ given $\mathbf{X}$ via Algorihm 1 (or its ADMM version, Algorithm 7, in the appendix). Let the obtained optimal solutions be $\mathbf{C}^*, \mathbf{b}^*$, and $\boldsymbol{\theta}^*$. For a given missing entry, we can use $\boldsymbol{\theta}^*$ only if all other features in the row of that missing entry are available. However, that is not usually the case in practice, as each row can contain more than one missing entry. Therefore, one can learn a separate model for each missing pattern in the dataset. Let us clarify this point through the example in Figure 1. In this example, we have three different missing patterns (one missing pattern for each dataset). For missing entries in Dataset 1, the first forty features are available. Let $\mathbf{r}_j$ denote the vector of the first 40 features in row $j$. Assume that we aim to impute entry $i \in \{41, \ldots, 100\}$ in row $j$ where $i$ denoted by $x_{ji}$. To this end, we restrict $\mathbf{X}$ to the first 40 features. Moreover, we consider $y = x_i$ as the target variable. Then, we run Algorithm 1 on $\mathbf{X}$ and $y$ to obtain the optimal $\mathbf{C}^*, \mathbf{b}_i^*$, and $\boldsymbol{\theta}_i^*$. Consequently, we impute $x_{ji}$ as follows:

$$x_{ji} = \mathbf{r}_j^T \boldsymbol{\theta}_i^*$$

We can use the same methodology for imputing missing entries in each feature for missing patterns in Dataset 2 and Dataset 3. While this approach is reasonable for the missing pattern observed in Figure 1, in many practical problems, different rows can have distinct missing patterns. Thus, in the worst case, Algorithm 1 must be executed once for each missing entry. Such an approach is computationally expensive and might be infeasible in large-scale datasets containing large amounts of missing entries. Alternatively, one can perform Algorithm 1 only once to obtain $\mathbf{C}^*$ and $\mathbf{b}^*$ (considered the "worst-case/pessimistic" estimation of the moments). Then to impute each missing entry, $\mathbf{C}^*$ and $\mathbf{b}^*$ are restricted to the features available in that missing entry's row. Having the restricted $\mathbf{C}^*$ and $\mathbf{b}^*$, the regressor $\boldsymbol{\theta}^*$ can be obtained in closed-form (line 6 in Algorithm 1). In this approach, we perform algorithm 1 once and find the optimal $\boldsymbol{\theta}^*$ for each missing entry based on the estimated $\mathbf{C}^*$ and $\mathbf{b}^*$. This approach can lead to sub-optimal solutions compared to the former approach, but it is much faster and more scalable.

**Beyond Linear Regression:** While the developed methods are primarily designed for ridge linear regression, one can apply non-linear transformations (kernels) to obtain models beyond linear. In Appendix E, we show how to extend the developed algorithms to quadratic models. The RIFLE framework applied to the quadratically transformed data is called **QRIFLE**.

## 4 Robust Classification Framework

In this section, we study the proposed framework in (4) for the classification tasks in the presence of missing values. Since the target variable $y \in \mathcal{Y} = \{1, \ldots, M\}$ takes discrete values in classification tasks, we consider the uncertainty sets over the data's first- and second-order marginals given each target value (label) separately. Therefore, the distributionally robust classification over low-order marginals can be described as:

$$
\begin{aligned}
\min_{\mathbf{w}} \max_{P} \quad & \mathbb{E}_P[\ell(\mathbf{x}, y, \mathbf{w})] \\
\text{s.t.} \quad & \boldsymbol{\mu}_{\min,y} \leq \mathbb{E}_P[\mathbf{x}|y] \leq \boldsymbol{\mu}_{\max,y} \quad \forall y \in \mathcal{Y} \\
& \boldsymbol{\Sigma}_{\min,y} \leq \mathbb{E}_P[\mathbf{x}\mathbf{x}^T|y] \leq \boldsymbol{\Sigma}_{\max,y} \quad \forall y \in \mathcal{Y}
\end{aligned}
\tag{11}
$$

where $\boldsymbol{\mu}_{\min}, \boldsymbol{\mu}_{\max}, \boldsymbol{\Sigma}_{\min}$, and $\boldsymbol{\Sigma}_{\max}$ are the estimated confidence intervals for the first and second order of the data distribution. Unlike the robust linear regression task in Section 3, the evaluation of the objective function in (11) might depend on higher-order marginals (beyond second-order) due to the nonlinearity of the loss function. As a result, Problem (11) is a non-convex non-concave intractable min-max optimization problem in general. For the sake of computational traceability, we restrict the distribution in the inner maximization problem to the set of normal distributions. In the following section, we specialize (11) to the quadratic discriminant analysis as a case study. The methodology can be extended to other popular classification algorithms, such as support vector machines and multi-layer neural networks.

### 4.1 Robust Quadratic Discriminant Analysis

Learning a logistic regression model on datasets containing missing values has been studied extensively in the literature (Fung & Wrobel, 1989; Abonazel & Ibrahim, 2018). Besides deleting missing values and imputation-based approaches, Fung & Wrobel (1989) models the logistic regression task in the presence of missing values as a linear discriminant analysis problem where the underlying assumption is that the predictors follow normal distribution conditional on the labels. Mathematically speaking, they assume that the data points assigned to a specific label follow a Gaussian distribution, i.e., $\mathbf{x}|y = i \sim N(\boldsymbol{\mu}_i, \boldsymbol{\Sigma})$. They use the available data to estimate the parameters of each Gaussian distribution. Therefore, the parameters of the logistic regression model can be assigned based on the estimated parameters of the Gaussian distributions for different classes. Similar to the linear regression case, the estimations of means and covariances are unbiased only when the data satisfies the MCAR condition. Moreover, when the number of data points in the dataset is small, the variance of the estimations can be very high. Thus, to train a logistic regression model that is robust to the percentage and different types of missing values, we specialize the general robust classification framework formulated in Equation (11) to the logistic regression model. Instead of considering a common covariance matrix for the conditional distributions of $\mathbf{x}$ given labels $y$ (linear discriminant analysis), we assume a more general case where each conditional distribution has its own covariance matrix (quadratic discriminant analysis). Assume that $\mathbf{x}|y \sim N(\boldsymbol{\mu}_y, \boldsymbol{\Sigma}_y)$ for $y = 0, 1$. We aim to find the optimal solution to the following problem:

$$
\begin{aligned}
\min_{\mathbf{w}} \max_{\boldsymbol{\mu}_0, \boldsymbol{\mu}_1, \boldsymbol{\Sigma}_0, \boldsymbol{\Sigma}_1} \quad & \mathbb{E}_{\mathbf{x}|y=1 \sim N(\boldsymbol{\mu}_1, \boldsymbol{\Sigma}_1)}\Big[ -\log\big(\sigma(\mathbf{w}^T\mathbf{x})\big)\Big]\mathbb{P}(y=1) + \\
& \mathbb{E}_{\mathbf{x}|y=0 \sim N(\boldsymbol{\mu}_0, \boldsymbol{\Sigma}_0)}\Big[ -\log\big(1 - \sigma(\mathbf{w}^T\mathbf{x})\big)\Big]\mathbb{P}(y=0) \\
\text{s.t.} \quad & \boldsymbol{\mu}_{\min_0} \leq \boldsymbol{\mu}_0 \leq \boldsymbol{\mu}_{\max_0} \\
& \boldsymbol{\mu}_{\min_1} \leq \boldsymbol{\mu}_1 \leq \boldsymbol{\mu}_{\max_1} \\
& \boldsymbol{\Sigma}_{\min_0} \leq \boldsymbol{\Sigma}_0 \leq \boldsymbol{\Sigma}_{\max_0} \\
& \boldsymbol{\Sigma}_{\min_1} \leq \boldsymbol{\Sigma}_1 \leq \boldsymbol{\Sigma}_{\max_1}
\end{aligned}
\tag{12}
$$

Where $\sigma(\mathbf{x}) = 1/\big(1 + \exp(-\mathbf{x})\big)$ is the sigmoid function.

To solve Problem (12), first, we focus on the scenario when the target variable has no missing values. In this case, each data point contributes to the estimation of either $(\boldsymbol{\mu}_1, \boldsymbol{\Sigma}_1)$ or $(\boldsymbol{\mu}_0, \boldsymbol{\Sigma}_0)$, depending on its label. Similar to the robust linear regression case, we can apply Algorithm 4 to estimate the confidence intervals for $\boldsymbol{\mu}_i, \boldsymbol{\Sigma}_i$ using data points whose target variable equals $i$ ($y = i$).

Obviously, the objective function is convex in $\mathbf{w}$ since the logistic regression loss is convex, and the expectation of loss can be seen as a weighted summation, which is convex. Thus, fixing $\boldsymbol{\mu}, \boldsymbol{\Sigma}$ the outer minimization problem can be solved with respect to $\mathbf{w}$ using standard first-order methods such as gradient descent.

Although the robust reformulation of logistic regression stated in (12) is convex in $\mathbf{w}$ and concave in $\boldsymbol{\mu}_0$ and $\boldsymbol{\mu}_1$, the inner maximization problem is intractable with respect to $\boldsymbol{\Sigma}_0$ and $\boldsymbol{\Sigma}_1$. We approximate Problem (12) in the following manner:

$$
\begin{aligned}
\min_{\mathbf{w}} \quad \max_{\boldsymbol{\mu}_0, \boldsymbol{\Sigma}_0, \boldsymbol{\mu}_1, \boldsymbol{\Sigma}_1} \quad & \pi_1 \mathbb{E}_{\mathbf{x}|y=1 \sim N(\boldsymbol{\mu}_1, \boldsymbol{\Sigma}_1)} \Big[ -\log \Big( \sigma(\mathbf{w}^T \mathbf{x}) \Big) \Big] + \pi_0 \mathbb{E}_{\mathbf{x}|y=0 \sim N(\boldsymbol{\mu}_0, \boldsymbol{\Sigma}_0)} \Big[ -\log \Big( 1 - \sigma(\mathbf{w}^T \mathbf{x}) \Big) \Big], \\
\text{s.t.} \quad & \boldsymbol{\mu}_{\min_0} \leq \boldsymbol{\mu}_0 \leq \boldsymbol{\mu}_{\max_0} \\
& \boldsymbol{\mu}_{\min_1} \leq \boldsymbol{\mu}_1 \leq \boldsymbol{\mu}_{\max_1} \\
& \boldsymbol{\Sigma}_0 \in \{\boldsymbol{\Sigma}_{01}, \boldsymbol{\Sigma}_{02}, \ldots, \boldsymbol{\Sigma}_{0k}\} \\
& \boldsymbol{\Sigma}_1 \in \{\boldsymbol{\Sigma}_{11}, \boldsymbol{\Sigma}_{12}, \ldots, \boldsymbol{\Sigma}_{1k}\},
\end{aligned}
\tag{13}
$$

where $\pi_1 = \mathbb{P}(y = 1)$ and $\pi_0 = \mathbb{P}(y = 0)$. To compute optimal $\boldsymbol{\mu}_0$ and $\boldsymbol{\mu}_1$, we have:

$$
\max_{\boldsymbol{\mu}_1} \quad \mathbb{E}_{\mathbf{x} \sim N(\boldsymbol{\mu}_1, \boldsymbol{\Sigma}_1)} \Big[ -\log \Big( \sigma(\mathbf{w}^T \mathbf{x}) \Big) \Big] \quad \text{s.t.} \quad \boldsymbol{\mu}_{\min} \leq \boldsymbol{\mu}_1 \leq \boldsymbol{\mu}_{\max}
\tag{14}
$$

**Theorem 3.** *Let $\mathbf{a}[i]$ be the $i$-th element of vector $\mathbf{a}$. The optimal solution of Problem (14) has the following form:*

$$
\boldsymbol{\mu}_1^*[i] = \begin{cases} \boldsymbol{\mu}_{\max}[i], & if \ \mathbf{w}[i] \leq 0 \\ \boldsymbol{\mu}_{\min}[i], & if \ \mathbf{w}[i] > 0. \end{cases}
\tag{15}
$$

Note that we relaxed (12) by taking the maximization problem over a finite set of $\boldsymbol{\Sigma}$ estimations. We estimate each $\boldsymbol{\Sigma}$ by bootstrapping on the available data using Algorithm 4. Define $f_i(\mathbf{w})$ as:

$$
f_i(\mathbf{w}) = \pi_1 \mathbb{E}_{\mathbf{x} \sim N(\boldsymbol{\mu}_1^*, \boldsymbol{\Sigma}_{i1})} \Big[ -\log \Big( \sigma(\mathbf{w}^T \mathbf{x}) \Big) \Big]
\tag{16}
$$

Similarly, we can define:

$$
g_i(\mathbf{w}) = \pi_0 \mathbb{E}_{\mathbf{x} \sim N(\boldsymbol{\mu}_0^*, \boldsymbol{\Sigma}_{i0})} \Big[ -\log \Big( 1 - \sigma(\mathbf{w}^T \mathbf{x}) \Big) \Big]
\tag{17}
$$

Since the maximization problem is over a finite set, we can rewrite Problem (13) as:

$$
\begin{aligned}
\min_{\mathbf{w}} \max_{i,j \in \{1,\ldots,k\}} f_i(\mathbf{w}) + g_j(\mathbf{w}) = \min_{\mathbf{w}} \max_{p_1,\ldots,p_k,q_1,\ldots,q_k} \quad & \sum_{i=1}^{k} p_i f_i(\mathbf{w}) + \sum_{j=1}^{k} p_i g_j(\mathbf{w}) \\
\text{s.t.} \quad & \sum_{i=1}^{k} p_i = 1, \quad p_i \geq 0 \\
& \sum_{j=1}^{k} q_j = 1, \quad q_j \geq 0
\end{aligned}
\tag{18}
$$

Since the maximum of several functions is not necessarily smooth (differentiable), we add a quadratic regularization term to the maximization problem, accelerating the convergence rate (Nouiehed et al., 2019) as follows:

$$
\begin{aligned}
\min_{\mathbf{w}} \max_{p_1,\ldots,p_k,q_1,\ldots,q_k} \quad & \sum_{i=1}^{k} p_i f_i(\mathbf{w}) - \delta \sum_{i=1}^{k} p_i^2 + \sum_{j=1}^{k} q_j g_j(\mathbf{w}) - \delta \sum_{j=1}^{k} q_j^2 \\
\text{s.t.} \quad & \sum_{i=1}^{k} p_i = 1, \quad p_i \geq 0 \\
& \sum_{j=1}^{k} q_j = 1, \quad q_j \geq 0
\end{aligned}
\tag{19}
$$

First, we show how to solve the inner maximization problem. Note that the $p_i$'s and $q_i$'s are independent. We show how to find optimal $p_i$'s. Optimizing with respect to $q_i$'s is similar. Since the maximization problem is a constrained quadratic program, we can write the Lagrangian function as follows:

$$\max_{p_1,\ldots,p_k} \sum_{i=1}^{k} p_i f_i(\mathbf{w}) - \delta \sum_{i=1}^{k} p_i^2 - \lambda(\sum_{i=1}^{k} p_i - 1) \tag{20}$$
$$\text{s.t.} \quad p_i \geq 0$$

Having the optimal $\lambda$, the above problem has a closed-form solution with respect to each $p_i$, which can be written as:

$$p_i^* = \left[ \frac{-\lambda + f_i}{2\delta} \right]_+$$

Since $p_i^*$ is a non-increasing function with respect to $\lambda$, we can find the optimal value of $\lambda$ using the following bisection algorithm. Algorithm 2 demonstrates how to find an $\epsilon$-optimal $\lambda$ and $p_i^*$'s efficiently using the bisection idea.

---

**Algorithm 2** Finding the optimal $\lambda$ and $p_i$'s using the bisection idea

---

1: **Initialize:** $\lambda_{\text{low}} = 0, \lambda_{\text{high}} = \max_i f_i, p_i = 0 \quad \forall i \in \{1, 2, \ldots, k\}$.
2: **while** $|\sum_{i=1}^{n} p_k - 1| > \epsilon$ **do**
3: $\quad \lambda = \frac{\lambda_{\text{low}} + \lambda_{\text{high}}}{2}$
4: $\quad$ Set $p_i = [\frac{-\lambda + f_i}{2\delta}]_+ \quad \forall i \in \{1, 2, \ldots, k\}$
5: $\quad$ **if** $\sum_{i=1}^{k} p_i < 1$ **then**
6: $\quad\quad \lambda_{\text{high}} = \lambda$
7: $\quad$ **else**
8: $\quad\quad \lambda_{\text{low}} = \lambda$
9: **return** $\lambda, p_1, p_2, \ldots, p_k$.

---

**Remark 4.** *An alternative method for finding optimal $\lambda$, and $p_i$'s is to sort $f_i$ values in $\mathcal{O}(k \log k)$ first, and then finding the smallest $f_i$ such that if we set $\lambda = f_i$, the sum of $p_i$'s is bigger than 1 (let $j$ be the index of that value). Without loss of generality, assume that $f_1 \leq \cdots \leq f_k$. Then, $\sum_{i=j}^{k} \frac{-\lambda + f_i}{2\delta} = 1$, which has a closed-form solution with respect to $\lambda$.*

To update $\mathbf{w}$, we need to solve the following optimization problem:

$$\min_{\mathbf{w}} \sum_{i=1}^{k} p_i^* f_i(\mathbf{w}) + \sum_{j=1}^{k} q_j^* g_i(\mathbf{w}), \tag{21}$$

Similar to the standard statistical learning framework, we solve the following empirical risk minimization problem by applying the gradient descent to $\mathbf{w}$ on a finite data sample. Define $\hat{f}_i$ as follows:

$$\hat{f}_i(\mathbf{w}) = \pi_1 \sum_{t=1}^{n} \left[ -\log\left(\sigma(\mathbf{w}^T \mathbf{x}_t)\right) \right], \tag{22}$$

where $\mathbf{x}_1, \ldots, \mathbf{x}_n$ are generated from the distribution $\mathcal{N}(\boldsymbol{\mu}_1^*, \boldsymbol{\Sigma}_{1i})$. The empirical risk minimization problem can be written as follows:

$$\min_{\mathbf{w}} \sum_{i=1}^{k} p_i^* \hat{f}_i(\mathbf{w}) + \sum_{j=1}^{k} q_j^* \hat{g}_i(\mathbf{w}), \tag{23}$$

Algorithm 3 summarizes the robust linear discriminant analysis method for the case where the label of all data points is available. Theorem 5 demonstrates the convergence of gradient descent algorithm applied to (23) in $\mathcal{O}\left(\frac{k}{\epsilon} \log(\frac{M}{\epsilon})\right)$ iterations to an $\epsilon$-optimal solution.

---

**Algorithm 3** Robust Quadratic Discriminant Analysis in the Presence of Missing Values

---

1: **Input**: $\mathbf{X}_0, \mathbf{X}_1$: matrix of data points with labels 0 and 1 respectively, $T$ : Number of iterations, $\alpha$ : Step-size.
2: Estimate $\boldsymbol{\mu}_{\min_0}$ and $\boldsymbol{\mu}_{\max_0}$ using the available entries of $\mathbf{X}_0$.
3: Estimate $\boldsymbol{\mu}_{\min_1}$ and $\boldsymbol{\mu}_{\max_1}$ using the available entries of $\mathbf{X}_1$.
4: Estimate $\boldsymbol{\Sigma}_{01}, \ldots, \boldsymbol{\Sigma}_{0k}$ using bootstrap estimator on the available data of $\mathbf{X}_0$.
5: Estimate $\boldsymbol{\Sigma}_{11}, \ldots, \boldsymbol{\Sigma}_{1k}$ using bootstrap estimator on the available data of $\mathbf{X}_1$.
6: **for** $i = 1, \ldots, T$ **do**
7:     Compute $\boldsymbol{\mu}_1^*$ and $\boldsymbol{\mu}_0^*$ by Equation (15).
8:     Find optimal $p_1, \ldots, p_k$, and $q_1, \ldots, q_k$ using Algorithm 2.
9:     $\mathbf{w} = \mathbf{w} - \alpha \left( \sum_{i=1}^k p_i^* \nabla \hat{f}_i(\mathbf{w}) + \sum_{j=1}^k q_j^* \nabla \hat{g}_i(\mathbf{w}) \right)$

---

**Theorem 5.** *Assume that $M = \max_i f_i$. Gradient descent algorithm requires $\mathcal{O}\left( \frac{k}{\epsilon} \log(\frac{M}{\epsilon}) \right)$ gradient evaluations for converging to an $\epsilon$-optimal saddle point of the optimization problem* (23).

In Appendix F, we extend the methodology to the case where $y$ contains missing entries.

## 5 Experiments

In this section, we evaluate RIFLE's performance on a diverse set of inference tasks in the presence of missing values. We compare RIFLE's performance to several state-of-the-art approaches for data imputation on synthetic and real-world datasets. The experiments are designed in a manner that the sensitivity of the model to factors such as the number of samples, data dimension, types, and proportion of missing values can be evaluated. The description of all datasets used in the experiments can be found in Appendix I.

### 5.1 Evaluation Metrics

We need access to the ground-truth values of the missing entries to evaluate RIFLE and other state-of-the-art imputation approaches. Hence, we artificially mask a proportion of available data entries and predict them with different imputation methods. A method performs better than others if the predicted missing entries are closer to the ground-truth values. To measure the performance of RIFLE and the existing approaches on a regression task for a given test dataset consisting of $N$ data points, we use normalized root mean squared error (NRMSE), defined as:

$$\text{NRMSE} = \frac{\sqrt{\frac{1}{N} \sum_{i=1}^N (y_i - \hat{y}_i)^2}}{\sqrt{\frac{1}{N} \sum_{i=1}^N (y_i - \bar{y})^2}}$$

where $y_i$, $\hat{y}_i$, and $\bar{y}$ represent the true value of the $i$-th data point, the predicted value of the $i$-th data point, and the average of true values of data points, respectively. In all experiments, generated missing entries follow either a missing completely at random (MCAR) or a missing not at random (MNAR) pattern. A discussion on the procedure of generating these patterns can be found in Appendix G.

### 5.2 Tuning Hyper-parameters of RIFLE

The hyper-parameter $c$ in (7) controls the robustness of the model by adjusting the size of confidence intervals. This parameter is tuned by performing a cross-validation procedure over the set $\{0.1, 0.25, 0.5, 1, 2, 5, 10, 20, 50, 100\}$, and the one with the lowest NMRSE is chosen. The default value in the implementation is $c = 1$ since it consistently performs well over different experiments. Furthermore, $\lambda$, the hyper-parameter for the ridge regression regularizer, is tuned by choosing 20% of the data as the validation set from the set $\{0.01, 0.1, 0.5, 1, 2, 5, 10, 20, 50\}$. To tune $K$, the number of bootstrap samples for estimating the confidence intervals, we tried $10, 20, 50$, and $100$. No significant difference is observed in terms of the test performance for the above values.

Furthermore, we tune the hyper-parameters of the competing packages as follows. For KNN-Imputer (Troy-anskaya et al., 2001), we try $\{2, 10, 20, 50\}$ for the number of neighbors ($K$) and pick the one with the highest performance. For MICE (Buuren & Groothuis-Oudshoorn, 2010) and Amelia (Honaker et al., 2011), we generate 5 different imputed data and pick the one with the highest performance on the test data. MissForest has multiple hyper-parameters. We keep the criterion as "MSE" since our performance evaluation measure is NRMSE. Moreover, we tune the number of iterations and number of estimations (number of trees) by checking values from $\{5, 10, 20\}$ and $\{50, 100, 200\}$, respectively. We do not change the structure of the neural networks for MIDA (Gondara & Wang, 2018) and GAIN (Yoon et al., 2018), and the default versions are performed for imputing datasets.

### 5.3 RIFLE Consistency

In Theroem 2 Part (a), we demonstrated that RIFLE is consistent. In Figure 3, we investigate the consistency

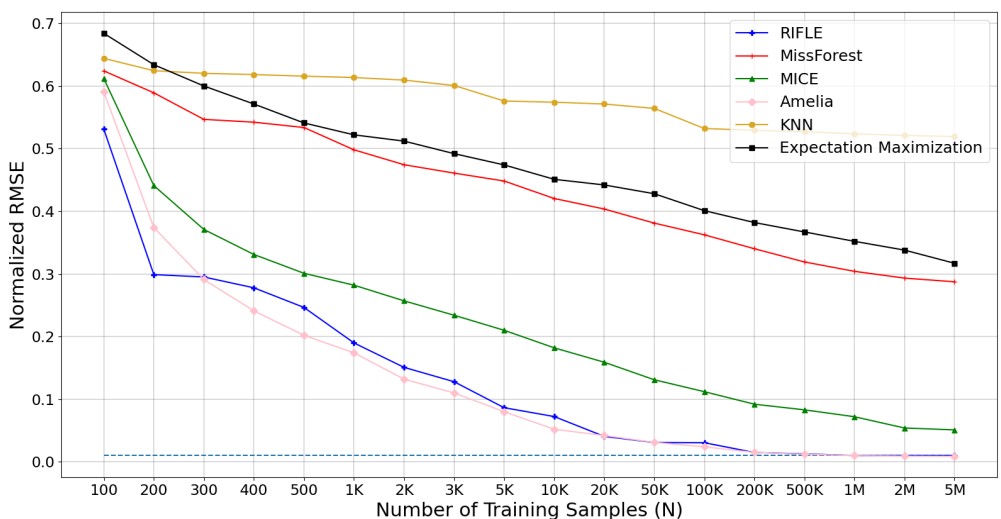

Figure 3: Comparing the consistency of RIFLE, MissForest, KNN Imputer, MICE, Amelia, and Expectation Maximization methods on a synthetic dataset containing 40% of missing values.

of RIFLE on synthetic datasets with different proportions of missing values. The synthetic data has 50 input features following a jointly normal distribution with the mean whose entries are randomly chosen from the interval $(-100, 100)$. Moreover, the covariance matrix equals $\Sigma = SS^T$ where $S$ elements are randomly picked from $(-1, 1)$. The dimension of $S$ is $50 \times 20$. The target variable is a linear function of input features added to a mean zero normal noise with a standard deviation of 0.01. As depicted in Figure 3, RIFLE requires fewer samples to recover the ground-truth parameters of the model compared to MissForest, KNN Imputer, Expectation Maximization (Dempster et al., 1977), and MICE. Amelia's performance is significantly good since the predictors have a joint normal distribution and the linear underlying model. Note that by increasing the number of samples, the NRMSE of our framework converges to 0.01, which is the standard deviation of the zero-mean Gaussian noise added to each target value (the dashed line).

### 5.4 Data Imputation via RIFLE

As explained in Section 3, while the primary goal of RIFLE is to learn a robust regression model in the presence of missing values, it can also be used as an imputation tool. We run RIFLE and several state-of-the-art approaches on five datasets from the UCI repository (Dua & Graff, 2017) (Spam, Housing, Clouds, Breast Cancer, and Parkinson datasets) with different proportions of MCAR missing values (the description of the datasets can be found in Appendix I). Then, we compute the NMRSE of imputed entries. Table 1 shows the performance of RIFLE compared to other approaches for the datasets where the proportion of missing values are relatively high $\left(\frac{n(1-p)}{d} \approx \mathcal{O}(1)\right)$. RIFLE outperforms these methods in almost all cases and performs slightly better than MissForest, which uses a highly non-linear model (random forest) to impute missing values.

| Dataset Name | RIFLE | QRIFLE | MICE | Amelia | GAIN | MissForest | MIDA | EM |
|---|---|---|---|---|---|---|---|---|
| Spam (30%) | 0.87 ±0.009 | **0.82** ±0.009 | 1.23 ±0.012 | 1.26 ±0.007 | 0.91 ±0.005 | 0.90 ±0.013 | 0.97 ±0.008 | 0.94 ± 0.004 |
| Spam (50%) | 0.90 ±0.013 | **0.86** ±0.014 | 1.29 ±0.018 | 1.33 ±0.024 | 0.93 ±0.015 | 0.92 ±0.011 | 0.99 ±0.011 | 0.97 ± 0.008 |
| Spam (70%) | 0.92 ±0.017 | **0.91** ±0.019 | 1.32 ±0.028 | 1.37 ±0.032 | 0.97 ±0.014 | 0.95 ±0.016 | 0.99 ±0.018 | 0.98 ± 0.017 |
| Housing (30%) | 0.86 ±0.015 | 0.89 ±0.018 | 1.03 ±0.024 | 1.02 ±0.016 | **0.82** ±0.015 | 0.84 ±0.018 | 0.93 ±0.025 | 0.95 ± 0.011 |
| Housing (50%) | **0.88** ±0.021 | 0.90 ±0.024 | 1.14 ±0.029 | 1.09 ±0.027 | **0.88** ±0.019 | **0.88** ±0.018 | 0.98 ±0.029 | 0.96 ± 0.016 |
| Housing (70%) | **0.92** ±0.026 | 0.95 ±0.028 | 1.22 ±0.036 | 1.18 ±0.034 | 0.95 ±0.027 | 0.93 ±0.024 | 1.02 ±0.037 | 0.98 ± 0.017 |
| Clouds (30%) | 0.81 ±0.018 | 0.79 ±0.019 | 0.98 ±0.024 | 1.04 ±0.027 | 0.76 ±0.021 | **0.71** ±0.011 | 0.83 ±0.022 | 0.86 ± 0.013 |
| Clouds (50%) | 0.84 ±0.026 | 0.84 ±0.028 | 1.10 ±0.041 | 1.13 ±0.046 | 0.82 ±0.027 | **0.75** ±0.023 | 0.88 ±0.033 | 0.89 ± 0.018 |
| Clouds (70%) | 0.87 ±0.029 | 0.90 ±0.033 | 1.16 ±0.044 | 1.19 ±0.048 | 0.89 ±0.035 | **0.81** ±0.031 | 0.93 ±0.044 | 0.92 ± 0.023 |
| Breast Cancer (30%) | **0.52** ±0.023 | 0.54 ±0.027 | 0.74 ±0.031 | 0.81 ±0.032 | 0.58 ±0.024 | 0.55 ±0.016 | 0.70 ±0.026 | 0.67 ± 0.014 |
| Breast Cancer (50%) | **0.56** ±0.026 | 0.59 ±0.027 | 0.79 ±0.029 | 0.85 ±0.033 | 0.64 ±0.025 | 0.59 ±0.022 | 0.76 ±0.035 | 0.69 ± 0.022 |
| Breast Cancer (70%) | **0.59** ±0.031 | 0.65 ±0.034 | 0.86 ±0.042 | 0.92 ±0.044 | 0.70 ±0.037 | 0.63 ±0.028 | 0.82 ±0.035 | 0.67 ± 0.014 |
| Parkinson (30%) | 0.57 ±0.016 | 0.55 ±0.016 | 0.71 ±0.019 | 0.67 ±0.021 | **0.53** ±0.015 | 0.54 ±0.010 | 0.62 ±0.017 | 0.64 ± 0.011 |
| Parkinson (50%) | 0.62 ±0.022 | 0.64 ±0.025 | 0.77 ±0.029 | 0.74 ±0.034 | **0.61** ±0.022 | 0.65 ±0.014 | 0.71 ±0.027 | 0.69 ± 0.022 |
| Parkinson (70%) | **0.67** ±0.027 | 0.74 ±0.033 | 0.85 ±0.038 | 0.82 ±0.037 | 0.69 ±0.031 | 0.73 ±0.022 | 0.78 ±0.038 | 0.75 ± 0.029 |

Table 1: Performance comparison of RIFLE, QRIFLE (Quadratic RIFLE), and state-of-the-art methods on several UCI datasets. We applied to impute methods on three different missing-value proportions for each dataset. The best imputer is highlighted with bold font, and the second-best imputer is underlined. Each experiment is done 5 times, and the average and the standard deviation of performances are reported.

## 5.5 Sensitivity of RIFLE to the Number of Samples and Proportion of Missing Values

In this section, we analyze the sensitivity of RIFLE and other state-of-the-art approaches to the number of samples and the proportion of missing values. In the experiment in Figure 4, we create 5 datasets containing

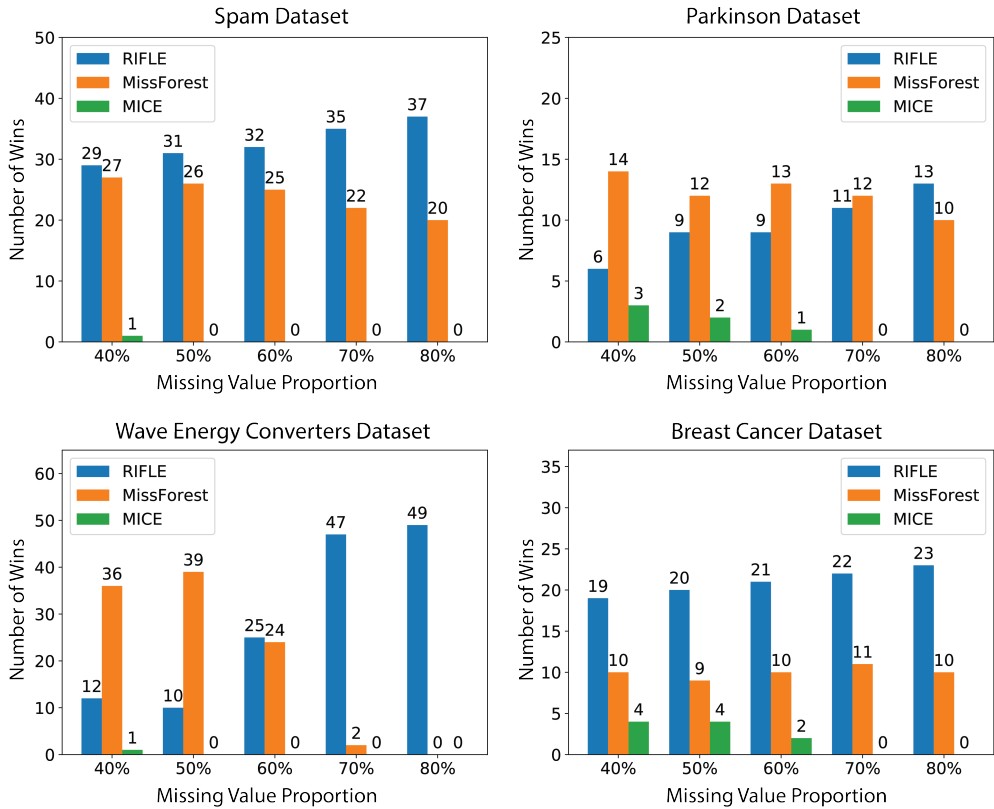

Figure 4: Performance Comparison of RIFLE, MICE, and MissForest on four UCI datasets: Parkinson, Spam, Wave Energy Converter, and Breast Cancer. For each dataset, we count the number of features that each method outperforms the others.

40%, 50%, 60%, 70%, and 80% of MCAR missing values, respectively, for four real datasets (Spam, Parkinson, Wave Energy Converter, and Breast Cancer) from UCI Repository (Dua & Graff, 2017) (the description of the datasets can be found in Appendix I). Given a feature in a dataset containing missing values, we say an imputer wins that feature if the imputation error in terms of NRMSE for that imputer is less than the error

of the other imputers. Figure 4 reports the number of features won by each imputer on the created datasets described above. As we observe, the number of wins for RIFLE increases as we increase the proportion of missing values. This observation shows that the sensitivity of RIFLE as an imputer to the proportion of missing values is less than MissForest and MICE in general.

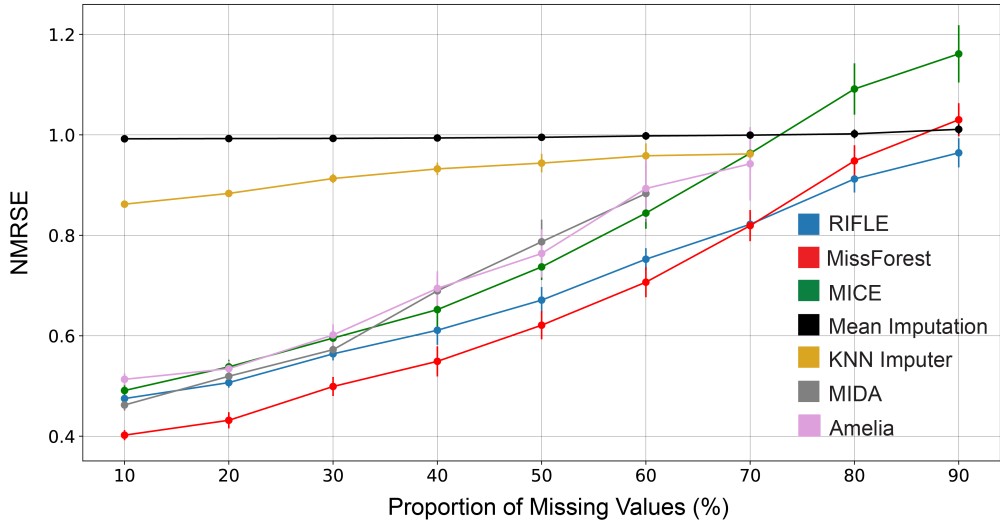

Figure 5: Sensitivity of RIFLE, MissForest, Amelia, KNN Imputer, MIDA, and Mean Imputer to the percentage of missing values on the Drive dataset. Increasing the percentage of missing value entries degrades the benchmarks' performance compared to RIFLE. KNN-imputer implementation cannot be executed on datasets containing 80% (or more) missing entries. Moreover, Amelia and MIDA do not converge to a solution when the percentage of missing value entries is higher than 70%.

Figure 4 does not show how the increase in the proportion of missing values changes the NRMSE of imputers. Next, we analyze the sensitivity of RIFLE and several imputers to change in missing value proportions. Fixing the proportion of missing values, we generate 10 random datasets containing missing values in random locations on the Drive dataset (the description of datasets is available in Appendix I). We impute the missing values for each dataset with RIFLE, MissForest, Mean Imputation, and MICE. Figure 5 shows the average and the standard deviation of these 4 imputers' performances for different proportions of missing values (10% to 90%). Figure 5 depicts the sensitivity of MissForest and RIFLE to the proportion of missing values in the Drive dataset. We select 400 data points for each experiment with different proportions of missing values (from 10% to 90%) and report the average NRMSE of imputed entries. Finally, in Figure 6, we have evaluated RIFLE and other methods on the BlogFeedback dataset (see Appendix I) containing 40% missing values. The results show that RIFLE's performance is less sensitive to decreasing the number of samples.

## 5.6 Performance Comparison on Real Datasets

In this section, we compare the performance of RIFLE to several state-of-the-art approaches, including MICE (Buuren & Groothuis-Oudshoorn, 2010), Amelia (Honaker et al., 2011), MissForest (Stekhoven & Bühlmann, 2012), KNN Imputer (Raghunathan et al., 2001), and MIDA (Gondara & Wang, 2018). There are two primary ways to do this. One method to predict a continuous target variable in a dataset with many missing values is first to impute the missing data with a state-of-the-art package, then run a linear regression. An alternative approach is to directly learn the target variable, as we discussed in Section 3.

Table 2 compares the performance of mean imputation, MICE, MIDA, MissForest, and KNN to that of RIFLE on three datasets: NHANES, Blog Feedback, and superconductivity. Both Blog Feedback and Superconductivity datasets contain 30% of MNAR missing values generated by Algorithm 9, with 10000 and 20000 training samples, respectively. The description of the NHANES data and its distribution of missing values can be found in Appendix I.

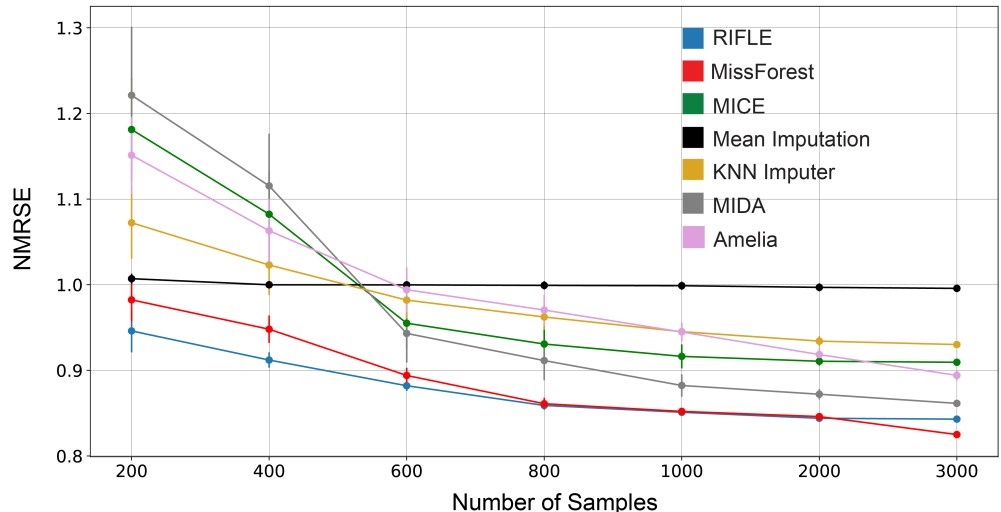

Figure 6: Sensitivity of RIFLE, MissForest, MICE, Amelia, Mean Imputer, KNN Imputer, and MIDA to the number of samples for the imputations of Blog Feedback dataset containing 40% of MCAR missing values. When the number of samples is limited, RIFLE outperforms other methods, and its performance is very close to the non-linear imputer MissForest for larger samples.

**Efficiency of RIFLE:** We perform RIFLE for 1000 iterations and the step size of 0.01 in the above experiments. At each iteration, the main operation is to find the optimal $\boldsymbol{\theta}$ for any given $\mathbf{b}$ and $\mathbf{C}$. The average time of each method on each dataset is reported in Table 5 in Appendix L. The main reason for the time efficiency of RIFLE compared to MICE, MissForest, MIDA, and KNN Imputer is that it directly predicts the target variable without imputation of all missing entries.

| Methods | Datasets | | |
|---|---|---|---|
| | Super Conductivity | Blog Feedback | NHANES |
| Regression on Complete Data | 0.4601 | 0.7432 | 0.6287 |
| **RIFLE** | $\mathbf{0.4873} \pm 0.0036$ | $0.8326 \pm 0.0085$ | $\mathbf{0.6304} \pm 0.0027$ |
| Mean Imputer + Regression | $0.6114 \pm 0.0006$ | $0.9235 \pm 0.0003$ | $0.6329 \pm 0.0008$ |
| MICE + Regression | $0.5078 \pm 0.0124$ | $0.8507 \pm 0.0325$ | $0.6612 \pm 0.0282$ |
| EM + Regression | $0.5172 \pm 0.0162$ | $0.8631 \pm 0.0117$ | $0.6392 \pm 0.0122$ |
| MIDA Imputer + Regression | $0.5213 \pm 0.0274$ | $0.8394 \pm 0.0342$ | $0.6542 \pm 0.0164$ |
| MissForest | $0.4925 \pm 0.0073$ | $\mathbf{0.8191} \pm 0.0083$ | $0.6365 \pm 0.0094$ |
| KNN Imputer | $0.5438 \pm 0.0193$ | $0.8828 \pm 0.0124$ | $0.6427 \pm 0.0135$ |

Table 2: Normalized RMSE of RIFLE and several state-of-the-art Methods on Superconductivity, blog feedback, and NHANES datasets. The first two datasets contain 30% Missing Not At Random (MNAR) missing values in the training phase generated by Algorithm 9. Each method applied 5 times to each dataset, and the result is reported as the average performance $\pm$ standard deviation of experiments in terms of NRMSE.

Since MICE and MIDA cannot predict values during the test phase without data imputation, we use them in a pre-processing stage to impute the data. Then we apply the linear regression to the imputed dataset. On the other hand, RIFLE, KNN imputer, and MissForest can predict the target variable without imputing the training dataset. Table 2 shows that RIFLE outperforms all other state-of-the-art approaches executed on the three mentioned datasets. In particular, RIFLE outperforms MissForest, while the underlying model RIFLE uses is simpler (linear) compared to the nonlinear random forest model utilized by Missforest.

| Number of Training Data Points | Method | | |
|:---:|:---:|:---:|:---:|
| | LDA | Robust LDA | Robust QDA |
| 50 | $52.38\% \pm 3.91\%$ | $62.14\% \pm 1.78\%$ | $61.36\% \pm 1.62\%$ |
| 100 | $61.24\% \pm 1.89\%$ | $68.46\% \pm 1.04\%$ | $70.07\% \pm 0.95\%$ |
| 200 | $73.49\% \pm 0.97\%$ | $73.35\% \pm 0.67\%$ | $73.51\% \pm 0.52\%$ |

Table 3: Sensitivity of Linear Discriminant Analysis, Robust LDA (Common Covariance Matrices), and Robust QDA (Different Covariance matrices for two groups) to the number of training samples.

### 5.6.1 Performance of RIFLE on Classification Tasks

In Section 4, we discussed how to specialize RIFLE to robust normal discriminant analysis in the presence of missing values. Since the maximization problem over the second moments of the data ($\mathbf{\Sigma}$) is intractable, we solved the maximization problem over a set of $k$ covariance matrices estimated by bootstrap sampling. To investigate the effect of choosing $k$ on the performance of the robust classifier, we train robust normal discriminant analysis models for different values of $k$ on two training datasets (Avila and Magic) containing 40% MCAR missing values. The description of the datasets can be found in Appendix I. For $k = 1$, there is no maximization problem, and thus, it is equivalent to the classifier proposed in Fung & Wrobel (1989). As shown in Figure 7, increasing the number of covariance estimations generally enhances the accuracy of the classifier in the test phase. However, as shown in Theorem 5, the required time for completing the training phase grows linearly regarding the number of covariance estimations.

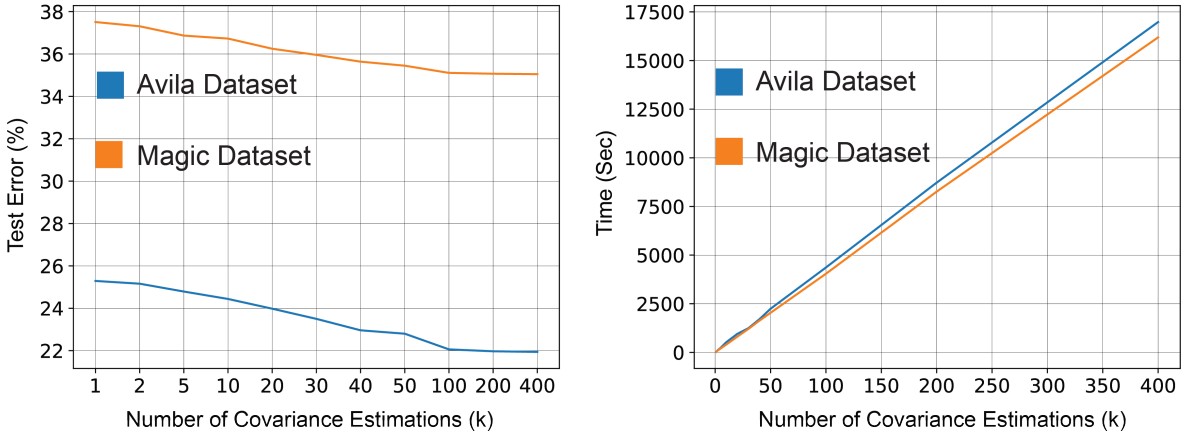

Figure 7: Effect of the number of covariance estimations on the performance (left) and run time (right) of robust LDA on Avila and Magic datasets. Increasing the number of covariance estimations ($k$) improves the model's accuracy on the test data. However, it takes longer training time.

### 5.6.2 Comparison of Robust Linear Regression and Robust QDA

An alternative approach to the robust QDA presented in Section 4 is to apply the robust linear regression algorithm (Section 3) and mapping the solutions to each one of the classes by thresholding (positive value maps to Label 1 and negative values to label $-1$).

Table 4 compares the performance of two classifiers on three different datasets. As demonstrated in the table, when all features are continuous, quadratic discriminant analysis has a better performance. It shows the QDA model relies highly on the normality assumption, while robust linear regression handles the categorical features better than robust QDA.

**Limitations and Future Directions:** The proposed framework for robust regression in the presence of missing values is limited to linear models. While in Appendix E, we use polynomial kernels to apply non-linear transformations on the data, such an approach can potentially increase the number of missing values in the

| Dataset | Feature Type | Accuracy of Methods | | | | | |
|---------|-------------|--------|-----------|-----------|------|-------------|-----|
| | | RIFLE | Robust QDA | MissForest | MICE | KNN Imputer | EM |
| Glass Identification | Continuous | $67.12\% \pm 1.84\%$ | $\mathbf{69.54\% \pm 1.97\%}$ | $65.76\% \pm 1.49\%$ | $62.48\% \pm 2.45\%$ | $60.37\% + \pm1.12\%$ | $68.21\% + \pm0.94\%$ |
| Annealing | Mixed | $63.41\% \pm 2.44\%$ | $59.51\% \pm 2.21\%$ | $\mathbf{64.91\% \pm 1.35}\%$ | $60.66\% \pm 1.59\%$ | $57.44\% \pm 1.44\%$ | $59.43\% + \pm1.29\%$ |
| Abalone | Mixed | $68.41\% \pm 0.74\%$ | $63.27\% \pm 0.76\%$ | $\mathbf{69.40\% \pm 0.42}\%$ | $63.12\% \pm 0.98\%$ | $62.43\% \pm 0.38\%$ | $62.91\% + \pm0.37\%$ |
| Lymphography | Discrete | $\mathbf{66.32}\% \pm \mathbf{1.05}\%$ | $58.15\% \pm 1.21\%$ | $66.11\% \pm 0.94\%$ | $55.73\% \pm 1.24$ | $57.39\% \pm 0.88\%$ | $59.55\% + \pm0.68\%$ |
| Adult | Discrete | $\mathbf{72.42}\% \pm \mathbf{0.06}\%$ | $60.36\% \pm 0.08$ | $70.34\% \pm 0.03\%$ | $63.30\% \pm 0.14\%$ | $60.14\% \pm 0.00$ | $60.69\% + \pm0.01\%$ |

Table 4: Accuracy of RIFLE, MICE, KNN-Imputer, Expectation Maximization (EM), and Robust QDA on different discrete, mixed, and continuous datasets. Robust QDA can perform better than other methods when the input features are continuous, and the target variable is discrete. However, RIFLE results in higher accuracy in mixed and discrete settings.

kernel space generated by the composition of the original features. A future direction is to develop efficient algorithms for non-linear regression models such as multi-layer neural networks, decision tree regressors, gradient boosting regressors, and support vector regression models. In the case of robust classification, the methodology is extendable to any loss beyond quadratic discriminant analysis. Unlike the regression case, a limitation of the proposed method for robust classification is its reliance on the Gaussianity assumption of data distribution (conditioned on each data label). A natural extension is to assume the underlying data distribution follows a mixture of Gaussian distributions.

**Conclusion:** In this paper, we proposed a distributionally robust optimization framework over the distributions with the low-order marginals within the estimated confidence intervals for inference and imputation of datasets in the presence of missing values. We developed algorithms for regression and classification with convergence guarantees. The method's performance is evaluated on synthetic and real datasets with different numbers of samples, dimensions, missing value proportions, and types of missing values. In most experiments, RIFLE consistently outperforms other existing methods.

# Acknowledgments

This work was supported by the NIH/NSF Grant 1R01LM013315-01, the NSF CAREER Award CCF-2144985, and the AFOSR Young Investigator Program Award FA9550-22-1-0192.

