# OpenReview forum: "RIFLE: Imputation and Robust Inference from Low Order Marginals"
_TMLR — Accepted by TMLR_

### Review · Reviewer_XaU5 · 2023-07-09

**Summary Of Contributions:**

This paper proposes a framework for prediction with missing values that combines the bootstrap to estimate data distribution characteristics and distributionally robust optimization. Explicit algorithms are given for ridge regression and discriminant analysis with convergence guarantees when the data are missing completely at random. Experiments are done on datasets (mostly from the UCI machine learning repository) with a range of sizes and dimensions. The proposed algorithm outperforms baselines when there is a greater amount of missing data and when the data may be missing not at random.

**Audience:**

Yes

**Broader Impact Concerns:**

The authors do not include a broader impact statement, which I think is fine since the work is largely methodological. The authors do not discuss the limitations; I feel that including them would strengthen the work, even if it's just some discussion of future work.

**Claims And Evidence:**

Yes

**Requested Changes:**

I think some discussion or analysis on the computational burden of the algorithm is critical.

**Strengths And Weaknesses:**

## Strengths
1. The paper is clearly written and the approach is straightforward to understand.
2. The algorithm has good theoretical support; convergence guarantees are provided.
3. Experimental results show that RIFLE improves over baselines when there is more limited data.

## Weaknesses
1. I feel too much was relegated to the appendix; some discussion of the classification case in the main text and more experiments for the classification setting would strengthen the work.
2. There is no discussion of the computational burden of RIFLE, in particular compared to the baselines, which would be of importance to practitioners.

---

> ### Author Response · Authors · 2023-08-09
>
> Thank you for your feedback. We are glad that you found the quality of the writing, experimental results, and theoretical supports interesting and of high quality. Below, we respond to your comments:
>
> **Comment:** *I feel too much was relegated to the appendix; some discussion of the classification case in the main text and more experiments for the classification setting would strengthen the work.*
>
> **Response:** After careful reading and discussions with the authors, we came to the same conclusion as you and agree that the classification section requires more details in the main body. Therefore, we moved more details from the appendix to the main body to explain the robust quadratic discriminant analysis methodology. Furthermore, we moved the experiments on the classification tasks to the main body.
>
> **Comment:** *There is no discussion of the computational burden of RIFLE, in particular compared to the baselines, which would be of importance to practitioners.*
>
> **Response:** Thank you for pointing out this issue. We have provided the convergence rate for the RIFLE algorithms in the main body. This rate would clarify the number of needed iterations for convergence and can be compared to the baselines. Notice that not all baselines have computational convergence guarantees. For example, GAIN is based on a nonconvex generative adversarial learning framework, for which the iteration complexity/convergence rate is unknown in the general machine learning/optimization community.
> Having said that, we agree with you that the actual runtime was missing in the original submission. Therefore, we added a new subsection in the experiments about the time efficiency of the RIFLE. We reported the actual runtime of the RIFLE on real datasets alongside the baselines in a new table added to the appendix.
>
> **Comment:** *The authors do not include a broader impact statement, which I think is fine since the work is largely methodological. The authors do not discuss the limitations; I feel that including them would strengthen the work, even if it's just some discussion of future work.*
>
> **Response:** We added a new section before the conclusion on “Limitations and Furure Work,” discussing the limitations of the current paper (linearity assumption for regression and normality assumption for the classification). Further, we discuss potential future directions and extensions of the current work.

---

### Review · Reviewer_bvAX · 2023-07-25

**Summary Of Contributions:**

This paper develops a general framework for robust inference with missing data, by combining distributionally robust optimization and bootstrap confidence intervals. More specifically, the optimization problem is formulated with $\ell_\infty$ constraints on low-order moments, which are computed from the incomplete data with a bootstrap method. The paper illustrates the framework on two examples, ridge regression and quadratic discriminant analysis. The paper provides some theory and conducts performance comparisons through numerical experiments.

**Audience:**

Yes

**Claims And Evidence:**

No

**Requested Changes:**

(1) The authors should rework the theory to make it precise and understandable. Furthermore, stronger theoretical guarantees should be provided, e.g., to quantify the efficiency loss due to distributional robustness.

(2) Extensions to the MAR setting should be discussed.

**Strengths And Weaknesses:**

Strengths:

It seems novel to combine the ideas of distributionally robust optimization and statistical inference for low-order moments. While confidence intervals for individual imputed values may be too wide, the sufficient statistics (i.e., low-order moments) are usually easier to estimate.

Weaknesses:

The proposed framework has two major weaknesses:

(1) In principle, it can only deal with data missing completely at random (MCAR), which is too restrictive. There are a wide variety of methods, including those based on inverse probability weighting and multiple imputation, that can handle data missing at random (MAR), which are more realistic.

(2) The distributionally robust approach has a price to pay and inevitably incurs an efficiency loss. The theoretical guarantees developed by the paper are quite weak and not sufficient to understand the trade-off between robustness and efficiency.

Also, Theorems 1 and 2 are not mathematically rigorous and contain many incorrect or ambiguous statements. For example:

(1) Theorem 1: $\boldsymbol\theta^*(\mathbf{b},\mathbf{C})$ is not defined; it is defined in Appendix H, Lemma 11. The assumption on a constant $\tau$ is questionable in the ridge regression example when $\mathbf{b}$ tends to infinity.

(2) Theorem 2, part (a): the limiting statements (8) and (9) do not specify the mode of convergence (e.g., in probability or almost surely). Also, the statements “the estimator proposed in Algorithm 2 converges to its actual value” and “by increasing the number of samples, the size of confidence intervals goes to zero” are not mathematically rigorous.

(3) Theorem 2, part (b): the definition of $L_{\text{test}}(\boldsymbol\theta)$ is problematic; if it is minimized over $\boldsymbol\theta$, then it should not depend on $\boldsymbol\theta$.

---

> ### Author Response · Authors · 2023-08-09
> **Response to Reviewer bvAX**
>
> Thank you for your detailed and constructive feedback.  Below, we respond to your comments in order:
>
> **Comment**: *(1) In principle, it can only deal with data missing completely at random (MCAR), which is too restrictive. There are a wide variety of methods, including those based on inverse probability weighting and multiple imputation, that can handle data missing at random (MAR), which are more realistic.*
>
> **Response:** We use the MCAR assumption to prove consistency and have a lower bound on the test accuracy of the method. While some other methods such as Amelia or GAIN use MAR assumption in developing their methodology, no convergence guarantee or statistical guarantee is provided. As another example, MICE, as the leading method for multiple imputations, does not have theoretical guarantees similar to ours. We need the MCAR assumption to ensure the estimated confidence intervals contain the ground-truth low-order marginals If our confidence intervals contain the actual ground-truth low-order marginals, we still can apply our result..  Notice that while our analysis is for the MCAR setting, we performed several experiments on real and artificial datasets containing MNAR missing patterns. The experiments demonstrate the superiority of our approach compared to the baselines even in the MNAR setting.
>
> **Comment**: *(2) The distributionally robust approach has a price to pay and inevitably incurs an efficiency loss. The theoretical guarantees developed by the paper are quite weak and not sufficient to understand the trade-off between robustness and efficiency.*
>
> **Response:** Thank you for bringing up this important discussion. We assume the reviewer is referring to the computational efficiency of the method. We offered the iteration complexity of Algorithm 1, Algorithm 2, and ADMM algorithm in the main body and in the appendix. If we apply [non-robust] linear regression, it requires $d^3$ operations complexity. Alternatively, algorithm 1 requires $Td^3$ iterations to converge where the iteration complexity for T is given in Theorem 1.  Therefore, the efficiency cost of the robust algorithm is T times the non-robust version. T is further improved using the Acceleration Method of Nesterov. Alternatively, one can solve the non-robust version of the algorithm using gradient descent in $O(\tilde{T} d^2 )$ computational complexity, where $\tilde{T}$ is the number of required iterations for solving the non-robust version. In this setting, we also use gradient descent to solve our subproblem for updating $\theta$, and again we have an overhead of $T$ in our complexity. While we pay an extra factor of $T$ in our iteration complexity, the good news is that this factor does not grow with $d$, as first-order methods are dimension-free. Finally, we would like to mention that, while we pay this extra factor, we gain much more in our inference accuracy.  We added the discussion on comparing non-robust and robust algorithms as a remark in the paper.
>
>
>
> **Comment:** *The definition of $\theta^{*}$ is not presented in the theorem statement.*
>
> **Response:** Thanks for mentioning this. We added the definition to the theorem statement.
>
> **Comment:** *The assumption on $\tau$ is questionable in the ridge regression example when $b$ tends to infinity.*
>
> **Response:** $\tau$ is a parameter that the Lipschitz constant depends on it. In real-world scenarios, $b$ and $C$ are bounded. Further, in the implementation, we normalize the data at first (e.g., each feature is mean zero, unit variance). Therefore, the covariances of pairwise features are bounded by 1. Therefore, the assumption is practical in most cases (since the data is bounded), and we have $\tau \approx O(1/\lambda).$
>
> **Comment:** *Theorem 2, part (a): the limiting statements (8) and (9) do not specify the mode of convergence (e.g., in probability or almost surely). Also, the statements “the estimator proposed in Algorithm 2 converges to its actual value” and “by increasing the number of samples, the size of confidence intervals goes to zero” are not mathematically rigorous.*
>
> **Response:** Thank you for pointing out this issue. We rewrote the theorem to address your concerns.
>
> **Comment:** *(3) Theorem 2, part (b): the definition of $L_{test}$ is problematic.*
>
> **Response:** Thanks for mentioning the problem in the $L_{test}$ definition. Similar to $L_{train}$, it should be defined without the minimization term. We corrected the definition in the revised manuscript.

---

### Review · Reviewer_opwB · 2023-07-27

**Summary Of Contributions:**

The paper proposes a methodology, called RIFLE, to handle missing data in supervised learning. The principle idea, Equation 4, is a variant of distributionally robust risk minimization: the uncertainty set is parametrized by mean and covariance conditions. The paper concretely instantiates the framework in Equation 4 to linear model-based hypothesis classes (Section 3 and Section 4), and suggests that the framework can be generalized beyond the linear case. Experiments indicate that the methodology performs favorably compared to existing approaches that handle missing data in supervised learning. For instance, Figure 3 shows that RIFLE has high prediction accuracy even in the presence of missing data, and Figure 4 shows that RIFLE performs accurate imputation of missing values.

**Audience:**

Yes

**Claims And Evidence:**

Yes

**Requested Changes:**

The primary change I would like to see is about Section 3.4. Currently I don't understand how Algorithm 1 is used to impute missing values.

Some secondary changes include:
* Figure 3: change the x-axis to be a log scale since the values are not equally spaced in the linear scale. This is also a problem in Figure 6.
* Section 5.2 the equation pointed to by the hyperlink (11) doesn't contain the hyperparameter c
* Reorganize the information in Table 1. I personally find tables hard to read. Perhaps you can convert Table 1 into a boxplot.

Time-permitting / future work
* Do you have results for linear discriminant analysis? Section 4 explains the methodology, but I don't see numerical demonstrations for this methodology in the main text.

**Strengths And Weaknesses:**

# Strengths

Missing data is a prevalent problem, so any methodology to address it has the potential to make a big impact. The proposed methodology is easy to understand, has only a small number of hyperparameters with clear meaning, and the paper adequately discusses ways to set such hyperparameters.

The writing is of high quality. Claims are supported by evidence. Sections/paragraphs have thesis statements that guide readers on what to take away. The transition between different sections is smooth.

The figures are very legible: the caption and the types of plots are clear in meaning, and corroborate with the text.

The proofs in the appendix are easy to verify.

# Weaknesses

The section on how RIFLE can be used to impute missing values (Section 3.4) is not as strong as the rest of the paper. I believe this section will be strengthened with more exposition. The current paper already utilizes Figure 1 - would it be possible to walk through a concrete example using this Figure 1, with more (but not too many) equations?

---

> ### Author Response · Authors · 2023-08-09
> **Response To Reviewer opwB**
>
> Thank you for your constructive feedback.   It is a pleasure to hear that you found the methodology, theoretical supports, experimental results, and writing quality compelling. We agree with your comments on the points that the paper can be improved on. Below we summarize what we did to address your concerns:
>
>
> **Comment**: *The primary change I would like to see is about Section 3.4. Currently, I don't understand how Algorithm 1 is used to impute missing values.*
>
> **Response**: We rewrote Section 3.4  and included a detailed procedure for using Algorithm 1 to impute data. We use Figure 1 as an example to explain how Algorithm 1 can impute the missing entries in Dataset 1. Further, when there are many distinct missing patterns in dataset rows, we suggest applying Algorithm1 once to obtain $C^*$ and $b^*$, and for each missing entry, finding the optimal $\theta$ by restricting $C^*$ and $b^*$ to the features that are available in the row of that missing entry. In other words, in the second approach, we run algorithm 1 once instead of running it for each missing entry pattern. Please see the revised section 3.4 for more details.
>
> **Comment**: *Time-permitting / future work: Do you have results for linear discriminant analysis?  Section 4 explains the methodology, but I don't see numerical demonstrations for this methodology in the main text.*
>
> **Response**: In our original submission, many of the explanations and the experiments were in the appendix. In our revised manuscript, we moved the details from the appendix to the main body to explain the robust quadratic discriminant analysis methodology. Further, we moved all experiments regarding the classification tasks to the main text.
>
> **Comment**:  *Some secondary changes include:
> Figure 3: change the x-axis to be a log scale since the values are not equally spaced in the linear scale. This is also a problem in Figure 6.
> Section 5.2 the equation pointed to by the hyperlink (11) doesn't contain the hyperparameter c
> Reorganize the information in Table 1. I personally find tables hard to read. Perhaps you can convert Table 1 into a boxplot.*
>
>
> **Response**: We changed the equation pointing to hyperlink (11) to the equation pointing to (7) (correcting the typo). Also, we tried the log scale for Figure 3. Unfortunately, in the log scale, the changes for the smaller numbers are sudden, and it is not easy to see the performance of different algorithms for 50, 100, and 200 number of samples. Furthermore, in Table 1, we report the performance of different algorithms for 15 different datasets. Plotting all of them in a box plot requires 15 different plots where the boundary of many boxes has large overlaps. It might be even harder to interpret the results in that case.

---

### Decision · Action_Editors · 2023-08-30

**Recommendation:** Accept as is

**Comment:**

The reviewers all agree that the submission provides a strong contribution to the general framework for robust inference with missing data. The claims are supported and all the concerns are well addressed. Thus I recommend acceptance.

**Audience:**

Yes

**Claims And Evidence:**

The claims are supported by accurate, convincing and clear evidence.

---

> ### Author Response · Authors · 2023-09-12
> **Camera-ready Version**
>
> Thank you and all the reviewers for their constructive feedback that helped us improve the paper's theory and experiments. We have uploaded the camera-ready version of our paper.